# Effect of muscle length on maximum evoked torque, discomfort, contraction fatigue, and strength adaptations during electrical stimulation in adult populations: A systematic review

Jonathan Galvão Tenório Cavalcante[1], Victor Hugo de Souza Ribeiro[2], Rita de Cássia Marqueti[3], Isabel de Almeida Paz[4], Júlia Aguillar Ivo Bastos[2], Marco Aurélio Vaz[4], Nicolas Babault[5], João Luiz Quagliotti Durigan[1,2]*

1 Laboratory of Muscle and Tendon Plasticity, Graduate Program of Physical Education, University of Brasília, Brasília, Distrito Federal, Brazil, 2 Laboratory of Muscle and Tendon Plasticity, Graduate Program of Rehabilitation Sciences, University of Brasília, Brasília, Distrito Federal, Brazil, 3 Molecular Analysis Laboratory, Graduate Program of Rehabilitation Sciences, University of Brasília, Brasília, Distrito Federal, Brazil, 4 Exercise Research Laboratory, School of Physical Education, Physical Therapy and Dance, Federal University of Rio Grande do Sul, Porto Alegre, RS, Brazil, 5 Centre d'Expertise de la Performance, INSERM UMR1093-CAPS, UFR des Sciences du Sport, University of Burgundy Franche-Comté, Besançon, Dijon, France

* durigan@unb.br

## Abstract

Neuromuscular electrical stimulation (NMES) can improve physical function in different populations. NMES-related outcomes may be influenced by muscle length (i.e., joint angle), a modulator of the force generation capacity of muscle fibers. Nevertheless, to date, there is no comprehensive synthesis of the available scientific evidence regarding the optimal joint angle for maximizing the effectiveness of NMES. We performed a systematic review to investigate the effect of muscle length on NMES-induced torque, discomfort, contraction fatigue, and strength training adaptations in healthy and clinical adult populations (PROSPERO: CRD42022332965). We conducted searches across seven electronic databases: PUBMED, Web of Science, EMBASE, PEDro, BIREME, SCIELO, and Cochrane, over the period from June 2022 to October 2023, without restricting the publication year. We included cross-sectional and longitudinal studies that used NMES as an intervention or assessment tool for comparing muscle lengths in adult populations. We excluded studies on vocalization, respiratory, or pelvic floor muscles. Data extraction was performed via a standardized form to gather information on participants, interventions, and outcomes. Risk of bias was assessed using the Revised Cochrane risk-of-bias tool for cross-over trials and the Physiotherapy Evidence Database scale. Out of the 1185 articles retrieved through our search strategy, we included 36 studies in our analysis, that included 448 healthy young participants (age: 19–40 years) in order to investigate maximum evoked torque (n = 268), contraction fatigability (n = 87), discomfort (n = 82), and muscle strengthening (n = 22), as well as six participants with spinal cord injuries, and 15 healthy older participants. Meta-analyses

**Data Availability Statement:** All relevant data are within the manuscript and its Supporting Information files.

**Funding:** J.G.T.C received a grant by Fundação de Apoio à Pesquisa do Distrito Federal (FAPDF) (grant number 00193-00001615/2023-00) for paying english language review and publication fee for the present study. FAPDF is a public research support foundation. https://www.fap.df.gov.br/.

**Competing interests:** The authors have declared that no competing interests exist.

were possible for comparing maximal evoked torque according to quadriceps muscle length through knee joint angle. At optimal muscle length 50˚ - 70˚ of knee flexion, where 0˚ is full extension), there was greater evoked torque during nerve stimulation compared to very short (0 – 30˚) ($p < 0.001$, CI 95%: -2.03, -1.15 for muscle belly stimulation, and -3.54, -1.16 for femoral nerve stimulation), short (31˚ - 49˚) ($p = 0.007$, CI 95%: -1.58, -0.25), and long (71˚ - 90˚) ($p < 0.001$, CI 95%: 0.29, 1.02) muscle lengths. At long muscle lengths, NMES evoked greater torque than very short ($p < 0.001$, CI 95%: -2.50, -0.67) and short ($p = 0.04$, CI 95%: -2.22, -0.06) lengths. The shortest quadriceps length generated the highest perceived discomfort for a given current amplitude. The amount of contraction fatigability was greater when muscle length allowed greater torque generation in the pre-fatigue condition. Strength gains were greater for a protocol at the optimal muscle length than for short muscle length. The quality of evidence was very high for most comparisons for evoked torque. However, further studies are necessary to achieve certainty for the other outcomes. Optimal muscle length should be considered the primary choice during NMES interventions, as it promotes higher levels of force production and may facilitate the preservation/gain in muscle force and mass, with reduced discomfort. However, a longer than optimal muscle length may also be used, due to possible muscle lengthening at high evoked tension. Thorough understanding of these physiological principles is imperative for the appropriate prescription of NMES for healthy and clinical populations.

## Introduction

Neuromuscular electrical stimulation (NMES) can improve neuromuscular function in different populations [1–3]. NMES-related outcomes may be influenced by several parameters, such as device-derived parameters (e.g., current type, current amplitude, stimulation frequency, pulse duration) [3–8], the device-human interface (e.g., electrode type, size, and configuration; nerve or muscle stimulation) [9,10], and human-derived parameters (e.g., type of contraction, target muscle, muscle-tendon unit length, health status) [11–17]. These factors should interact to produce greater evoked force, and speed up strengthening/hypertrophy while attenuating the perceived discomfort and contraction fatigability. Accordingly, muscle length is a widely recognized modulator of the force generation capacity of muscle fibers and the stimulus for strengthening and hypertrophy when it comes to voluntary exercise [18–20]. As such, the optimal muscle length for NMES, would be at the plateau region of the force-length relationship, obtained by manipulating joint angle [21–23]. However, to date, there are no summaries of the scientific data on how the NMES effects can be optimized by the "best joint angle", which may limit standardization of NMES-based programs and novel approaches to improve performance during NMES delivery, in addition to the long-term results.

The muscle force-length relationship of single muscle fibers *in vitro* is classically known to display an ascending and a descending limb, with a plateau in between, where the maximal tension is achieved [21], also supported by recent research [24]. *In vivo*, the joint angle-torque relationship becomes the practical translation of the force-length relationship for assessing human performance, taking into account that different moment arms, muscle-tendon unit architecture, and joint characteristics yield different shapes of the torque curve as the joint angle changes [11,15,22,25]. As such, the early study by Marsh et al. [26] brings relevant data showing the different shapes of the torque-angle relationship of the dorsiflexors according to

the type of activation (maximal voluntary contraction [MVC] or stimulated at different frequencies). However, while the dorsiflexors may show progressively greater evoked torque as the ankle angle moves towards plantar flexion [26], for the quadriceps muscle, the torque is usually greater closer to 60° of knee flexion [16], and for the biceps brachialis, the torque is usually greater close to 90° of elbow flexion [25]. These discrepancies in the torque-angle relationship for different joints reveal that this branch of knowledge must be broadened (for more joint angles and joint types) and systematically compiled so that researchers and clinicians can make evidence-based decisions when prescribing NMES.

When applying NMES, it is essential not only to consider torque generation but also to minimize the level of perceived discomfort. Forceful contractions at short muscle lengths (e.g., extended knee for knee extensor stimulation) can cause painful muscle cramps [1,27]. Therefore, during the application of NMES, in addition to biomechanical considerations, alterations in muscle length may contribute to varying levels of perceived discomfort. Thus, whenever possible, it is recommended that the NMES training be started at the 'ideal' muscle length (i.e., the joint angle that allows the greatest evoked torque generation) and that it subsequently progresses to isometric evoked contractions at longer muscle lengths. Despite the absence of evidence indicating that this approach diminishes discomfort during NMES sessions, it may mitigate the heightened risk of early muscle damage associated with training at longer muscle lengths [6,28].

A previous study found that NMES-induced quadriceps fatigability, i.e., the reduction in force generating capacity [29], depends on muscle length changes due to hip and knee joint angles [14], where fatigue is greater at a knee flexion of 60° compared to 20°. In the case of a 60° knee flexion, a supine position leads to earlier fatigue, which means greater performance fatigability [30], while with a 20° knee flexion, a supine position delays fatigue (here defined as a decline in an objective measure of performance over a discrete period of time) [11,30]. It has also been claimed that NMES at a shortened position (full knee extension) for strengthening the quadriceps promotes greater joint soft tissue protection after anterior cruciate ligament repair [31]. However, according to the authors, the strength gains observed were lower than expected, considering previous studies that used a knee flexion angle of 60° [31,32]. These results corroborate those of a pioneering study [33] that reported a greater strengthening effect for an elongated position (65° of knee flexion) compared to a shortened position (full knee extension). While abundant and emerging data exist to address the influence of muscle length on the benefits of NMES training programs, to date, no systematic reviews have identified the optimal muscle lengths for the generation of higher evoked torque with lower discomfort and contraction fatigability, and a greater strengthening/hypertrophy effect during NMES rehabilitation and training programs.

The current review, therefore, was developed to summarize the research comparing different muscle length settings during NMES, following the Cochrane collaboration recommendations [34] to assess the effects of these interventions on outcomes important for NMES-based programs. Specifically, we compared the effects of muscle length on NMES-evoked isometric torque, contraction fatigability, discomfort, and strength training adaptations in healthy and clinical adult populations. We hypothesized that the optimal muscle length, according to the preferable joint angle for maximum force development, would induce greater contraction fatigue, mainly due to the greater absolute muscle force (and torque) in the fresh (pre-fatigue) state. We also hypothesized that the discomfort would be lower at the optimal muscle length during NMES contractions. Finally, we hypothesized that strength gains would be more pronounced at the optimal muscle length, while greater hypertrophy could potentially be attained through more elongated positions in comparison to the optimal muscle length.

## Material and methods

This systematic review followed the recommendations proposed by the Cochrane Collaboration and the Preferred Reporting Items for Systematic Reviews and Meta-Analyses Statement (PRISMA) [34,35], and was registered at PROSPERO (CRD42022332965). S1 File.

### Search strategy

We searched seven electronic databases: PUBMED, Web Of Science, EMBASE, PEDro, BIREME, SCIELO, and Cochrane, from June 2022 to October 2023. The search strategy was established following the PICO strategy (Population, Intervention, Control, Outcome) for any adult population ($\geq$ 18 years old, healthy or clinical) submitted to electrical stimulation applied at different muscle lengths (according to joint angle) and that evaluated its effects on maximum evoked torque, contraction fatigability, perceived discomfort, or strength training adaptations.

We used the following descriptors in our search strategy, without restrictions on language or date of publication: "healthy", "adults", "participants", "volunteers". "muscle weakness", "muscle atrophy", "cachexia", "elderly", "muscle diseases", "muscle paralysis", "Parkinson", "neuromuscular disease", "stroke", "multiple sclerosis", "anterior cruciate ligament reconstruction", "chronic obstructive pulmonary disease", "lung disease", "cardiac disease", "obese", "vascular disease", "diabetes", "orthopedic patients", "nephrology patients", "electric stimulation", "functional electrical stimulation", "neuromuscular electrical stimulation", "muscle length", "joint angle", "joint position", "torque", "force", "fatigue", "discomfort", "pain". The searches were adapted for each database to identify all relevant articles. Additional articles were screened in the reference lists of included studies. The search strategy for each database is described in S2 File. The grey literature (available literature not published under a rigorous, peer-reviewed, independent scientific review system) was not searched, in order to avoid introducing bias and low-quality designs that could reduce the validity of our results.

The inclusion criteria were: i) cross-sectional, repeated measures, or randomized controlled trials; ii) comparisons of different muscle lengths (i.e., joint angles) during or after electrically induced muscle contraction as an intervention to generate maximum torque, contraction fatigability, perceived discomfort, or strength training adaptations; iii) healthy or clinical adult populations ($\geq$ 18 years old). Studies were excluded if they investigated the vocalization, respiratory, or pelvic floor muscles.

### Data extraction

Two researchers (J.G.T.C and V.H.R) independently evaluated the titles of all articles found using the search strategy. If the title was clearly on a topic not relevant for the present review, the article was excluded. In the case of uncertainty, the article was selected for abstract reading, along with those that could possibly be included. The same procedure was performed during abstract reading, i.e., studies whose abstracts did not provide sufficient information regarding the inclusion and exclusion criteria were selected for full-text evaluation. Disagreements among reviewers were resolved by consensus, and if conflict persisted, a third reviewer (J.L.Q. D.) was consulted. The data extraction was performed independently by the same two reviewers via a standardized form to gather information on participants (health status, age, anthropometrics), interventions (joint angle comparisons, electrode characteristics and placement, electrical current parameters), and outcomes. No automation tools were used.

For meta-analyses, when multiple data were available for maximum evoked torque measurements, we always chose the data expected to produce greater torque (e.g., doublet instead of twitch-evoked torque, or 50 Hz instead of 20 Hz of stimulation frequency), and we always

chose data not explicitly affected by previous stimulus (e.g., non-potentiated instead of potentiated). For contraction fatigue, we were interested in the evoked torque decline during a fatiguing protocol, as well as the evoked and or voluntary torque obtained before and after the fatiguing protocol. When the study results were only available in graphs, we extracted the data using ImageJ software (v. 1.46; National Institutes of Health, Bethesda, Maryland, USA).

## Risk of bias assessment

Risk of bias was assessed using the Revised Cochrane risk-of-bias tool for cross-over trials (RoB-CO) and the Physiotherapy Evidence Database (PEDro) scale. The RoB-CO assesses six domains: Bias arising from the randomization process; Bias arising from period and carryover effects; Bias due to deviations from intended intervention; Bias due to missing outcome data; Bias in measurement of the outcome; and Bias in selection of the reported result. The PEDro scale contains 11 items that involve: 1) eligibility criteria (not used to calculate score); 2) random allocation; 3) concealed allocation; 4) baseline comparability; 5) blinded subjects; 6) blinded therapists; 7) blinded assessors; 8) adequate follow-up; 9) intention-to-treat analysis; 10) between-group statistical comparisons; 11) point estimate and variability. Each item was marked as "yes (1/0)" or "no (0/0)" and scored on a 0 to 10 scale.

## Quality of evidence

The Grading of Recommendations, Assessment, Development, and Evaluation (GRADE) was used to assess the overall quality of evidence. The GRADE contains 5 domains: Study design and risk of bias; Inconsistency; Indirectness; Imprecision; and Other factors (e.g., reporting bias, publication bias). Regarding classification, high-quality of evidence means consistent results in at least 75% of the clinical trials of good methodological quality, presenting consistent, direct, and precise data, with no suspicious or known publication bias, and further research is unlikely to alter the estimate or the confidence in the results. Moderate quality of evidence means that at least one domain is not met, and new research is likely to significantly impact the confidence in the effect estimate. Low-quality evidence means that two of the domains are not met, and further research is expected to significantly impact the confidence in the effect estimate and is likely to alter the estimate. Very low-quality evidence means that three domains are not met, and the results are highly uncertain [36].

## Data analysis

Meta-analyses were performed using Software Review Manager 5.4.1 (The Cochrane Collaboration) if the available data were sufficient and at least two studies could be fairly compared, i.e., by matching the main characteristics: population, muscle involved, and muscle length. Considering these basic rules, there was compatibility for meta-analyses only for the evoked torque, and not for perceived discomfort, contraction fatigue, and strength training adaptations. The continuous values (mean and standard deviation) of maximum evoked torque, and the number of participants for each group comparison were extracted to estimate the standardized mean difference (SMD) of the intervention and its 95% confidence interval (CI) using a random-effects model with inverse variance as a statistical method. The test for overall effect (Z-test) provided the p-value. The level of significance was set at $p < 0.05$.

The reference to joint angle in the included studies can change; for example, the full knee extension is more commonly cited as 0˚, although it may also be considered as 180˚. Therefore, we defined full extension as 0˚ for the knee and hip joints. When discussing other joints, we provide the reference angle as needed. To enhance study compatibility for meta-analysis, we categorized the quadriceps muscle length (with a fixed hip angle set between 70˚ and 90˚)

based on the knee angle, as follows: very short (0˚ - 30˚), short (31˚ - 49˚), optimal (50˚ - 70˚), long (71˚ - 90˚), and very long (> 90˚). Most studies used round numbers to specify knee joint angles, although some studies involving the ankle and elbow joints employed specific angles (e.g., 16˚, 48˚, 104˚) [25,37].

A gold standard for classifying quadriceps muscle length is currently lacking. However, a range between 50˚ and 70˚, with 60˚ being the common midpoint, may be considered as optimal or intermediate length [38–41]. The terms 'short' and 'long' are typically defined within the context of the specific joint angles being compared. For instance, Skurvydas et al. [42] defined 'short' as a knee flexion angle of 50˚ and 'long' as 90˚, while Place et al. [43] used 'short' for 35˚ and 'long' for 75˚ of knee flexion. In contrast, Rassier [39] described 'optimal' as 60˚, 'shorter than optimal' as 30˚, and 'longer than optimal' as 90˚. When performing a meta-analysis, we ensured that the stimulated muscles, participant groups (comprising healthy young individuals), and methodology for torque generation (nerve or motor point stimulation) were consistent across the studies.

## Results

### Search findings

Our search strategy retrieved 1185 records. After removing 277 duplicates and 82 clinical trial registries, 826 studies were screened, together with six additional studies: one study found on a reference list [44], two found through a manual search on Google [45,46], two published by some of the authors of this systematic review [14,38], and one provided by an author upon request [47], totalizing 832 for title screening. Titles that were not eligible were excluded (n = 761). Thereafter, 71 studies were screened through the abstracts, with 42 studies remaining for full-text screening. After full-text reading, six studies were excluded; one standardized the same percentage of MVC during evoked torque (not allowing any comparison of maximum evoked torque), one did not assess isometric torque (torque values were obtained as the knee was moved through a range of motion that was not stated), one did not specify joint angles, one did not show any result that could allow joint angle/muscle length comparison, one contained the same variable and sample of another study by the same authors, and one was in the Japanese language. Consequently, the current review includes a total of 36 studies. The PRISMA flowchart of study identification and selection is presented in Fig 1.

### Quality of evidence

The results of the RoB-CO assessment can be found in S3 File. In brief, none of the studies exhibited bias resulting from deviations in the intended intervention, missing outcome data, outcome measurement issues, or selective result reporting. While one study drew attention to potential carryover effects, three studies did not provide clear reporting on their randomization process. Moreover, ten studies raised concerns as they presented a high risk or potential issues related to carryover effects. The GRADE assessments are provided along with the description of the meta-analyses results.

The results of the PEDro scale are shown in Table 1. Three studies scored '7', 22 studies scored '6', eight studies scored '5', and three studies scored '4'.

### Main characteristics of the included studies

Table 2 outlines the key characteristics of the studies included in our analysis. The studies were published between 1981 and 2022, written in English, and were categorized as employing a repeated measure design (crossover), randomized clinical trial, or self-control (where opposite

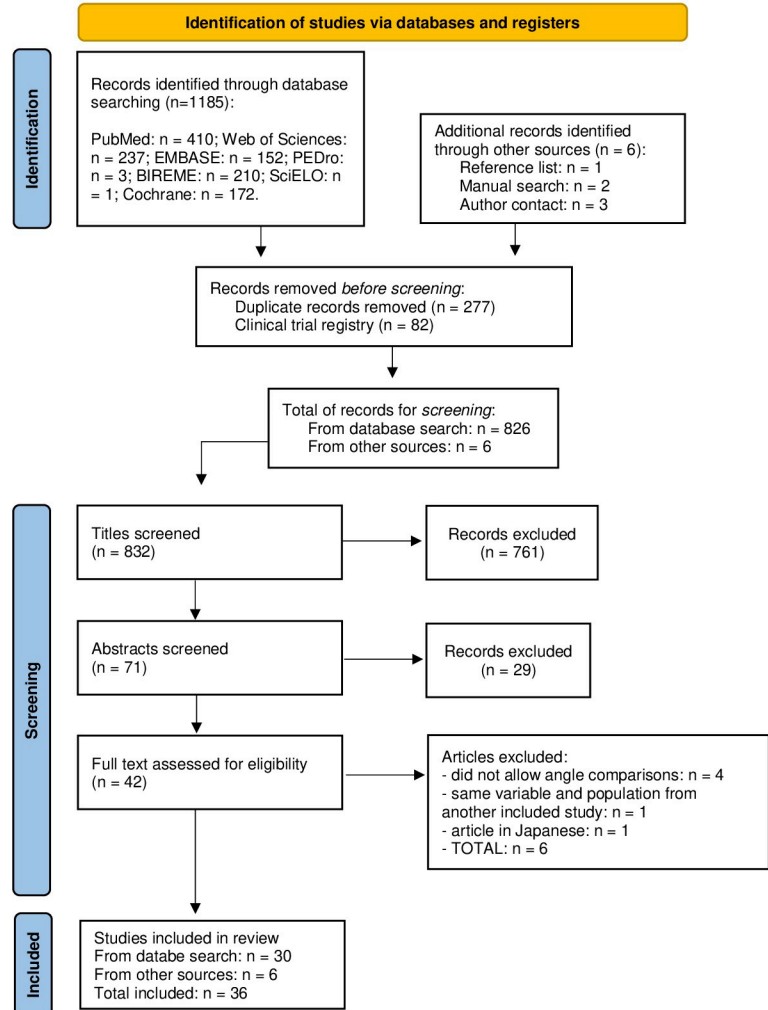

**Fig 1. Flowchart for identifying and selecting articles for final inclusion (based on the Prisma flowchart template).**

limbs were compared) approach. A total of 448 healthy young participants, ranging in age from 19 to 40 years, were enrolled across these studies. Their contributions were aimed at assessing various aspects, including maximum evoked torque (n = 268), contraction fatigability (n = 87), discomfort (n = 82), and muscle strengthening (n = 22). Moreover, six patients with spinal cord injuries were included and paired with an able-bodied sample [16], and one study dealt only with a healthy older sample of 15 participants [48]. In addition, 23 studies reported a measure of maximal electrically induced torque (peak twitch, supramaximal triplet stimulation, tetanic torque, or evoked torque), eight studies applied an NMES-fatiguing protocol, five studies reported the perceived discomfort, one study assessed strength gains [33], and two studies assessed other strength outcomes (without reporting only torque): the rate of torque development [49] and the extra forces with preceding activity, i.e., NMES-induced contractions through central pathways that lead to an increase in force increments that is disproportionate to the input/electrical current applied [50]. In this case, a 25 Hz-10 0Hz-25 Hz stimulation was applied at different ankle joints and the ratio of the torque generated in the second over the first 25 Hz stimuli was calculated.

**Table 1. Methodological quality of the included articles (PEDro scale).**

| Author (Year) | Random allocation | Concealed allocation | Similar groups baseline | Subject blinding | Therapist blinding | Assessor blinding | Adequate follow-up | Intention-to-treat analysis | Between-group comparisons | Point estimate & variability | Total |
|---|---|---|---|---|---|---|---|---|---|---|---|
| Cavalcante et al., 2022 [14] | Y | Y | Y | Y | N | N | Y | N | Y | Y | 7 |
| Cavalcante et al., 2021 [11] | Y | Y | Y | Y | N | N | Y | N | Y | Y | 7 |
| Harnie et al., 2020 [57] | Y | Y | Y | N | N | N | Y | N | Y | Y | 6 |
| Hali *et al.*, 2021 [74] | Y | Y | Y | N | N | N | N | N | Y | Y | 5 |
| Fouré et al., 2020 [28] | Y | Y | Y | N | N | N | N | N | Y | Y | 5 |
| Debenham & Power, 2019 [49] | N | N | Y | N | N | N | Y | N | Y | Y | 4 |
| Scott et al., 2019 [53] | Y | Y | Y | Y | N | N | Y | N | Y | Y | 7 |
| Gavin et al., 2018 [48] | Y | Y | Y | N | N | N | Y | N | Y | Y | 6 |
| Merlet et al., 2018 [76] | Y | Y | Y | N | N | N | Y | N | Y | Y | 6 |
| Yanase et al., 2017 [62] | Y | Y | Y | N | N | N | Y | N | Y | Y | 6 |
| Visscher et al., 2017 [71] | Y | Y | Y | N | N | N | Y | N | Y | Y | 6 |
| Bampouras et al., 2017 [65] | Y | Y | Y | N | N | N | N | N | Y | Y | 5 |
| Ando et al., 2018 [58] | Y | Y | Y | N | N | N | Y | N | Y | Y | 6 |
| Bremner et al., 2015a [45] | Y | Y | Y | N | N | N | Y | N | Y | Y | 6 |
| Bremner et al., 2015b [47] | Y | Y | Y | N | N | N | Y | N | Y | Y | 6 |
| Frigon et al., 2011 [50] | Y | Y | Y | N | N | N | Y | N | Y | Y | 6 |
| Skurvydas et al., 2010 [42] | Y | Y | Y | N | N | N | N | N | Y | Y | 5 |
| Marion et al., 2009 [60] | Y | Y | Y | N | N | N | Y | N | Y | Y | 6 |
| Ruiter et al., 2008 [52] | Y | Y | Y | N | N | N | Y | N | Y | Y | 6 |
| Kooistra et al., 2007 [54] | Y | Y | Y | N | N | N | Y | N | Y | Y | 6 |
| Lee et al., 2007 [59] | Y | Y | Y | N | N | N | Y | N | Y | Y | 6 |
| Gerrits et al., 2005 [16] | Y | Y | Y | N | N | N | Y | N | Y | Y | 6 |
| Miyamoto & Oda, 2005 [78] | Y | Y | Y | N | N | N | N | N | Y | Y | 5 |

*(Continued)*

**Table 1.** (Continued)

| Author (Year) | Random allocation | Concealed allocation | Similar groups baseline | Subject blinding | Therapist blinding | Assessor blinding | Adequate follow-up | Intention-to-treat analysis | Between-group comparisons | Point estimate & variability | Total |
|---|---|---|---|---|---|---|---|---|---|---|---|
| Kooistra et al., 2005 [56] | Y | Y | Y | N | N | N | N | N | Y | Y | 5 |
| Place et al., 2005 [43] | Y | Y | Y | N | N | N | Y | N | Y | Y | 6 |
| Ruiter et al., 2004 [55] | Y | Y | Y | N | N | N | Y | N | Y | Y | 6 |
| Babault et al., 2003 [38] | Y | Y | Y | N | N | N | Y | N | Y | Y | 6 |
| Maffiuletti et al., 2003 [72] | Y | Y | Y | N | N | N | Y | N | Y | Y | 6 |
| Hansen et al., 2003 [25] | Y | Y | Y | N | N | N | N | N | Y | Y | 5 |
| Mela et al., 2001 [37] | Y | Y | Y | N | N | N | N | N | Y | Y | 5 |
| Rassier, 2000 [39] | Y | Y | Y | N | N | N | Y | N | Y | Y | 6 |
| Sacco et al., 1994 [61] | N | N | Y | N | N | N | Y | N | Y | Y | 4 |
| McNeal & Bake, 1988 [46] | N | N | Y | N | N | N | Y | N | Y | Y | 4 |
| Fitch & McComas, 1985 [44] | Y | Y | Y | N | N | N | Y | N | Y | Y | 6 |
| Fahey et al., 1985 [33] | Y | Y | Y | N | N | N | Y | N | Y | Y | 6 |
| Marsh et al., 1981 [26] | Y | Y | Y | N | N | N | Y | N | Y | Y | 6 |

Y and grey marked: Yes; N and White marked: No.

## Physical parameters of the electrical stimulation protocols

All parameters used in the interventions are presented in Table 3. The current type and waveform were generally not fully described, and only two studies reported both parameters. Pulse duration ranged between 50 and 1000 μs, although 2 studies did not report pulse duration. When a tetanic contraction was applied in the evaluation and/or intervention, studies (n = 4) used stimulation ranging from 75 Hz to 300 Hz, with a pulse duration ranging from 200 μs to 600 μs. The exceptions were the two studies by Bremner et al. [45,47] that used a median frequency current (2500 Hz delivered in bursts of 50 Hz, also known as Russian current).

The stimulated muscles and joint angles are presented in Table 2. The quadriceps femoris muscle was the most commonly studied muscle, followed by the ankle plantar flexors and dorsiflexors, and the elbow flexors. Only one study investigated the external rotators (infraspinatus) of the shoulder.

## Evoked torque

Meta-analyses were only possible for this outcome. In total, 23 studies reported a measure of maximum evoked contraction through motor point stimulation (tetanic contraction or twitch)

**Table 2. Characteristics of the included studies.**

| Authors | Sample Size/ % male | Sample Characteristics | Electrode number (size), and type | Electrode Positions | Joint angles | Outcomes |
|---|---|---|---|---|---|---|
| **Cavalcante et al., 2022** [14] | 20/100% | Healthy; age: 24±4.6 y; body mass: 77±9.3 kg; height: 177.6 ±6.3 cm | 4 (5x5 cm) self-adhesive | Right quadriceps muscle bellies | Hip: 0° or 85° Knee: 20° or 60° | Contraction fatigue |
| **Cavalcante et al., 2021** [11] | 20/100% | Healthy; age: 24±4.6 y; body mass: 77±9.3 kg; height: 177.6 ±6.3 cm | 4 (5x5 cm) self-adhesive | Right quadriceps muscle bellies | Hip: 0° or 85° Knee: 20° or 60° | Evoked torque Discomfort |
| **Harnie et al., 2020** [57] | 32/72% | Healthy; age: 22.1±3.2 y; body mass: 67.5±10.3 kg; height: 174.7±8.2 cm | cathode (0.5 cm diameter) 1 (5x9 cm) | Right leg; femoral triangle, 3–5 cm below the inguinal ligament and gluteal fold | Hip: Knee: 90° or 30° | Peak twitch torque |
| **Hali *et al.*, 2021** [74] | 10/100% | Healthy; age: 24±3 y; weight: 81±7 kg; height: 181±5 cm | Not informed | Non-dominant leg (left), tibial nerve | Hip: 90° Knee: 90° Ankle: 20° DF or 20° PF | Peak twitch torque |
| **Fouré et al., 2020** [28] | 10/60% | Healthy; age: 27±4 y; body mass: 63.5±9.2 kg; height: 173 ±10 cm | 1 (5x10 cm) 2 (5x5 cm) | Proximal thigh: ~5cm below inguinal ligament) VL and VM muscle bellies | Hip: ~90° Knee: 50° or 100° | Contraction fatigue |
| **Debenham & Power, 2019** [49] | 8/100% | Healthy; 24 ± 3 y; body mass: 72±11kg; height: 177±9 cm | Anode: ECG electrode Cathode: (6–8x8–10 cm) aluminum electrode pad wrapped in a damp paper towel covered in conductive gel | Anode: inguinal triangle Cathode: inferior gluteal fold. | Hip: Not informed Knee: 35° and 100° | Rate of Torque Development |
| **Scott et al., 2019** [53] | 18/50% | Healthy; age: 24.7±5.9 y; body mass: 78.5±13.2 kg; height: 173.3±11 cm | 2 (7.62x12.7 cm) | Right proximal and distal quadriceps muscles | Hip: Seated Knee: 30, 60 or 90 | Evoked torque |
| **Gavin et al., 2018** [48] | 15/46.7% | Healthy old; 66 ± 8 y; body mass: 73.0 ± 14.1; height: 168.3 ± 8.2 cm | Single device | Common peroneal nerve (muscles: peroneus longus and tibialis anterior) | Hip: ~90° Knee: 0°, 45°, 90° | Discomfort |
| **Merlet et al., 2018** [76] | 12/58% | Healthy; age: 22.5±1.2 y; body mass: 63.5±9.2 kg; height: 172.5±9.7 cm | 1 (1.0 cm diameter) 1 (5x10 cm) | Right posterior tibial nerve | Hip: 70° Knee: 0°, 30°, 90° ankle: 90° | Peak twitch torque |
| **Yanase et al., 2017** [62] | 40/100% | Healthy; age 24.4 ± 3.6 y; body mass: 67.3 ± 7.6; height: 172.1 ± 5.6 cm | Bipolar electrodes, 5x5 cm | Right or left shoulder; one electrode over the motor point of the infraspinatus, other over the muscle belly. | Maximal internal rotation (IR; 82.5° ± 9.6°); neutral rotation (NEUT); max external rotation (ER; 86.1° ± 13.5°), | Muscle swelling and soreness |
| **Visscher et al., 2017** [71] | 16/100% | Healthy; age: 26±4 y; body mass: 78±6 kg; height: 182±5 cm | 1 (5 cm diameter) 1 (5x10 cm) | Femoral nerve trunk (individual muscle belly stimulation omitted) | Hip: 90° Knee: 30°, 65°, 90° | Peak twitch torque |
| **Bampouras et al., 2017** [65] | 9/100% | Healthy; age: 30.2±7.7 y; body mass: 81.7±11.2 kg; height: 178±0.09 cm | 2 (7x12.5 cm) | Proximal and distal regions of the quadriceps muscle group with the cathode being the proximal electrode. | Hip: 90° or 160° Knee: 90° | Tetanic torque |
| **Ando et al., 2018** [58] | 8/100% | Healthy; age: 24±2 y; body mass: 63.5±9.2 kg; height: 172.5±9.7 cm | 1 cathode (2.0x3.5 cm) and 2 anodes (7x10 cm) | Femoral nerve | Hips: 70° of flexion (0° is anatomical position) Knee: 60° (extended) and 110° (flexed) | Contraction fatigue |
| **Bremner et al., 2015a** [45] | 16/0% | Healthy; age: 21.5±2.4 y; body mass: 67.7±7.7 kg; height: 162.4±5.3 cm | 4 (5x9 cm) | Right leg; electrodes were placed over the femoral nerve and the motor points of each of the superficial quadriceps muscles | Hip: 85° Knee: 15° or 60° | Normalized peak torque |

(*Continued*)

**Table 2.** (Continued)

| Authors | Sample Size/ % male | Sample Characteristics | Electrode number (size), and type | Electrode Positions | Joint angles | Outcomes |
|---|---|---|---|---|---|---|
| **Bremner et al., 2015b** [47] | 20/50% | Healthy; age: 21.3 ± 2.1 y; body mass: 75.6 ± 15.3; height: 167.4 ± 8.5 cm | 4 (5x9 cm) | Over the proximal and distal VL, proximal RF, and distal VM motor points. | Hip: 85° Knee: 15° or 60° | Discomfort |
| **Frigon et al., 2011** [50] | 14/ 64.28% | Healthy; no other data available | TA: 5 cm, rounded; GM: 7x13 cm | Proximal and distal TA; and proximal and distal GM | Hip: 90° Knee: 170 – 180° (full extension) Ankle: | Extra forces |
| **Skurvydas et al., 2010** [42] | 11/100% | Healthy; 24.8 ± 3.7 y; body mass: 78.2 ± 4.7kg; height: 179.9 ± 3.6 cm | 2 carbonized rubber electrodes covered gel: 6x11 cm (proximal) and 6x20 cm (distal) | Right quadriceps; transversely across the width of the proximal and the distal portion of the quadriceps. | Hip: seated Knee: not informed Ankle: 9° dorsiflexion, 16° plantar flexion, and 44° plantar flexion | Evoked torque |
| **Marion et al., 2009** [60] | 8/70% | Healthy; age: 28.2±3.63 y | 2 (7.5x12.5 cm) | Right quadriceps muscle | Hips: 15° Knee: 20°, 40°, 65°, and 90° | Contraction fatigue |
| **Ruiter et al., 2008** [52] | 10/100% | Healthy; age: 26.0±7.1 y; body mass: 73.0±5.2 kg; height: 181.9±8.3 cm | Cathode: 5x5 cm Anode: 13x8 cm | Cathode: femoral nerve Anode: gluteal fold | Hip: 70° Knee: 30°, 60°, 90° | Triplet torque |
| **Kooistra et al., 2007** [54] | 7/100% | Healthy; age: 23–32 y; body mass: 69–83 kg; height: 173–193 cm | Cathode: 5x5 cm Anode: 13x8 cm | Cathode: femoral nerve Anode: gluteal fold | Hip: 70° Knee: 30°, 60°, 90° | Triplet torque |
| **Lee et al., 2007** [59] | 5/60% | Healthy; age: 22.5±1.2 y; body mass: 70.2±12 kg; height: 172.9±7.3 | - | Right quadriceps muscle | Hips:75° of flexion Knee: 15° and 90° | Contraction fatigue |
| **Gerrits et al., 2005** [16] | 8/- | Healthy; age: 37±4 y; body mass: 78±3 kg; height: 183±2 cm | 1 (5x5 cm) 1 (8x13 cm) | Femoral nerve (cathode) and medial part of the quadriceps muscle (anode) | Hip: 70° Knee: 30 to 90 (10° step) | Triplet torque |
| **Miyamoto & Oda, 2005** [78] | 9/100% | Healthy; age: 24.5±1.1 y; body mass: 68.1±2.6 kg; height: 171.3±2.8 cm | Carbon-impregnated rubber electrodes (3.5x2 cm) with conductive gel and secured to the right upper arm with adhesive tape. | One electrode was placed on the motor point of the biceps brachii, and the other was placed on the distal portion of the biceps brachii. | Elbow: 75, 90, 105, 120, 135, and 150 (full extension = 180). | 30 Hz Tetanic torque; Peak Twitch torque |
| **Kooistra et al., 2005** [56] | 8/100% | Healthy; age: 25.0±3.7 | Cathode: 5 x 5 cm Anode: 13 x 8 cm | Cathode: femoral nerve Anode: gluteal fold | Hip: 70° Knee: 30°, 60°, and 90° | Triplet torque |
| **Place et al., 2005** [43] | 11/100% | Healthy; age: 24±4 y; body mass: 67±7 kg; height: 177±6 cm | Cathode: monopolar ball electrode 0.5 cm diameter Anode: 10x5 cm | Cathode: femoral nerve Anode: gluteal fold | Hip: 90° Knee: 35°, 75° | Peak twitch torque |
| **Ruiter et al., 2004** [55] | 7/100% | Healthy; age: 19–40 y; body mass: 79–85 kg; height: 172–194 cm | Cathode: 5 x 5 cm Anode: 13 x 8 cm (?) | Cathode: femoral nerve Anode: gluteal fold | Hip: 70° (?) Knee: 30°, 60°, 90° | Octet |
| **Babault et al., 2003** [38] | 9/100% | Healthy; age: 20.6±1.6 y; body mass: 73.8±6.3 kg; height: 177.3±5.7 cm | Cathode: ball probe (100 mm diameter) Anode: 10 x 5 cm | Cathode: femoral nerve Anode: between the greater trochanter and the inferior iliac crest. | Hip: 90° Knee: 35°, 55°, 75° | Peak twitch torque |
| **Maffiuletti et al., 2003** [72] | 11/100% | Healthy; age: 26.4±5.4 y; body mass t: 71.4±8.8 kg; height: 177.5±6.8 cm | 1 (0.5 cm diameter) 1 (5 x 10 cm) | Right leg; femoral triangle, 3–5 cm below the inguinal ligament and gluteal fold | Hip: 0° or 90° Knee: 90° | Peak twitch torque |
| **Hansen et al., 2003** [25] | 13/ 53.8% | Healthy; 26.1±3.2 y; weight: 68.57±0.6 g; height: 173.9 ±8.7 cm | 2 electrodes; size not mentioned | Biceps brachialis motor point | Shoulder: not informed Elbow: 48, 62, 76, 90, 104, 118, 132, 146, and 160 | Doublet twitch torque |

(*Continued*)

**Table 2.** (Continued)

| Authors | Sample Size/ % male | Sample Characteristics | Electrode number (size), and type | Electrode Positions | Joint angles | Outcomes |
|---|---|---|---|---|---|---|
| **Mela et al., 2001** [37] | 6/66.6% | Healthy; age: 30.3±6.8 y | Cathode: circular (1.5-cm radius); indifferent: 5x9 cm | Cathode on deep peroneal nerve; indifferent electrode on distal bony part of the shank | Hip: seated<br>Knee: not informed<br>Ankle: 9˚ dorsiflexion, 16˚ plantar flexion, and 44˚ plantar flexion | Peak twitch torque |
| **Rassier, 2000** [39] | 10/- | Healthy; no other data available | 2 (13 x 12 cm) | Right quadriceps muscle (proximally and distally) | Hip: seated<br>Knee: 30˚, 60˚, 90˚" | Twitch torque Fatigue |
| **Sacco et al., 1994** [61] | 14/- | Healthy; age: 34 (25–59 y) | 2 (4-cm-diam) | Tibialis anterior | Knee: Not informed<br>Ankle: 100˚ (optimum) and 80˚ (short) | Contraction fatigue |
| **McNeal & Bake, 1988** [46] | 10/50% | Healthy; age: 23–33 y | 1 (4x5 cm)<br>1 (4x9 cm) | Left quadriceps muscle bellies | Hip: 60<br>Knee: 15˚, 45˚, 75˚ | Evoked torque |
| **Fitch & McComas, 1985** [44] | 10/- | Healthy; age: 31.1 ± 8.5 | rectangular pieces of aluminum foil (2x4 cm); | Cathode over peroneal nerve; anode over the anterior-superior aspect of tibialis anterior | 15˚ and 25˚ of dorsiflexion | Contraction fatigue |
| **Fahey et al., 1985** [33] | 55/ 50.90% | Healthy; females: age: 26.9 ± 4.27; body mass: 54.66 ± 5.19; height: 161.68 ± 6.21; males: age: 26.73 ± 3.52; body mass: 77.56 ± 7.24; height: 177.6 ± 7.06 | 2 (5.1x10.2 cm) | Quadriceps muscle bellies | Hip: unclear<br>Knee: 0˚ or 65˚ | Strengthening for isometric and isokinetic torque |
| **Marsh et al., 1981** [26] | 5/100% | Healthy; age 19–37 y | Cathode: oval lead plate 5x2 cm, covered by saline-soaked cloth); anode: cloth pad soaked in saline and mounted on a steel plate (8x8 cm) | Cathode: upper part of TA; Anode: the lower third of TA. | 11 angles (from 30˚ PF to 20˚ DF), in steps of 5˚ | Tetanic stimulation at 40 Hz |

DF: Dorsiflexion; PF: Plantar flexion; (?) presumed considering other articles by the authors; *.

or nerve stimulation (twitch, doublet, triplet, or octet) comparing the value reached at different muscle lengths. Of these, 17 were conducted on the knee extensors, two on the ankle plantar flexors, two on the ankle dorsiflexors, and two on the biceps brachialis muscle.

A total of seven meta-analyses were conducted, involving data from 13 studies (Fig 2A–2G) that applied NMES to the quadriceps femoris muscles (there were no matching studies for other muscle groups). Although the study by Gerrits et al. [16] met the criteria for inclusion in the meta-analyses, only the data from their able-bodied participants were used, as their sample also encompassed individuals with spinal cord injuries. Furthermore, according to the Cochrane recommendations [51], triplet torque data (sample, mean, and standard deviation) for knee flexion angles of 50˚, 60˚, and 70˚ were combined/collapsed to represent optimal (mid-range) muscle length, while the data for angles of 80˚ and 90˚ were combined to represent long muscle length. Collapsing data was also necessary in the study by Ruiter et al. [52], which compared the quadriceps triplet torque produced at 10˚, 30˚ 60˚, and 90˚ of knee flexion. In this case, 10˚ and 30˚ were collapsed to represent short muscle length. Notably, there was a significant effect (SMD: -1.59, CI 95%: -2.03, -1.15, p < 0.001) that favored optimal muscle length over very short muscle length during muscle belly stimulation (Fig 2A). This observation was drawn from three studies characterized by high-quality evidence [11,45,53].

**Table 3. NMES parameters used in the included studies.**

| Author | Current Type | Waveform | Pulse Frequency (Hz) | Phase Duration (µs) | Time ON/OFF (sec) or Duty Cycle | Additional parameters | Fatigue protocol duration | Intensity |
|---|---|---|---|---|---|---|---|---|
| Cavalcante et al., 2022 [14] | Bipolar | - | 100 | 500 | 10 s / 120 s | - | Does not apply | MEIC (VAS 8/10) |
| Cavalcante et al., 2021 [11] | Bipolar | - | 100 | 500 | 10 s / 120 s | - | 12 contractions | MEIC (VAS 8/10) |
| Harnie et al., 2020 [57] | Constant | Rectangular | Single pulse | 1000 | Does not apply | 400 V | Does not apply | 120% supramaximal (84.4 ±15.7 mA) |
| Hali et al., 2021 [74] | - | Square | Single pulse | 200 | Does not apply | 400 V | Does not apply | 20% supramaximal stimulation |
| Fouré et al., 2020 [28] | Biphasic symmetric | Rectangular | 100 | 400 | 5 s / 35 s | - | 40 contractions | MEIC |
| Debenham & Power, 2019 [49] | - | Square | 300 (Octet) | Single pulse: 1000; octet: 200 | - | 400 V | Does not apply | Single pulse: 15% above plateau<br>Octet: plateau (maximum torque) |
| Scott et al., 2019 [53] | - | Square | 75 | 600 | 4 s / 60 s | - | Does not apply | MEIC (VAS 7/10) |
| Gavin et al., 2018 [48] | - | - | Single pulse | 400* | - | - | Does not apply | 27 mA |
| Merlet et al., 2018 [76] | - | Rectangular | Single pulse | 1000 | - | 400 V | Does not apply | 120% twitch force |
| Yanase et al., 2017 [62] | - | - | 20 | 250 | 5 s / 2 s | - | 20 min (fatigue not assessed) | Maximum tolerated |
| Visscher et al., 2017 [71] | - | Rectangular | Single pulse | 1000 | - | - | Does not apply | 100 mA |
| Bampouras et al., 2017 [65] | Bipolar | - | 100 | 200 | 1 s/- | 10 ms IPI | Does not apply | Supramaximal (512±124.6 mA) |
| Ando et al., 2018 [58] | Constant | - | 20 | 200 | 70 s / 0 s | 400 V | 1 contraction | 120% twitch force (36–134 mA) |
| Bremner et al., 2015a [45] | Russian | | Carrier frequency:2500 | Not informed | / 120 s | Burst frequency: 50 bursts/s | Does not apply | Not clear |
| Bremner et al., 2015b [47] | Russian | - | Carrier frequency: 2500 | Not informed | 15 s (single contraction) | Burst frequency: 50 bursts/s | Does not apply | 30% - 40% MVC |
| Frigon et al., 2011 [50] | - | - | 25 and 100 | 1000 | - | - | Does not apply | 10–15% of Maximal evoked twitch at an ankle joint of 90˚ for plantar flexors, and at 120˚ for dorsiflexors |
| Skurvydas et al., 2010 [42] | - | Square | 100* | 500 | 1 s / 10 s | - | Does not apply | 10% supramaximal |
| Marion et al., 2009 [60] | Constant | - | 33 | 600 | 1 s / 1 s | - | 78 contractions | 20% MVC at 90˚ of knee flexion |
| Ruiter et al., 2008 [52] | Constant | - | 300 (triplet) | 200 | Does not apply | Does not apply | Does not apply | 50 mA over maximum at 90˚ of knee flexion (250–400 mA). |
| Kooistra et al., 2007 [54] | - | Rectangular | 300 (triplet) | 100 | Does not apply | Does not apply | Does not apply | 50 mA over maximum at each angle |
| Lee et al., 2007 [59] | - | - | 40 | 600 | Not clear | - | 120 contractions | 20% MVC |

*(Continued)*

**Table 3.** (Continued)

| Author | Current Type | Waveform | Pulse Frequency (Hz) | Phase Duration (µs) | Time ON/OFF (sec) or Duty Cycle | Additional parameters | Fatigue protocol duration | Intensity |
|---|---|---|---|---|---|---|---|---|
| Gerrits et al., 2005 [16] | - | Square | 300 (triplet) | 200 | / 120 s | - | Does not apply | Supramaximal |
| Miyamoto & Oda, 2005 [78] | - | Rectangular | 30* | 100 | - | - | Does not apply | maximal twitch contraction |
| Kooistra et al., 2005 [56] | - | Rectangular | 300 (triplet) | 100 | - | - | Does not apply | 50 mA over maximum torque (procedure at 60˚, but used for the other positions. i.e., same amplitude for all positions. |
| Place et al., 2005 [43] | Constant | - | Does not apply | 1000 | Does not apply | 400 V | Does not apply | Maximum (300–500 mA) |
| Ruiter et al., 2004 [55] | Constant | - | 300 | 100 | Does not apply | Does not apply | Does not apply | Maximum |
| Babault et al., 2003 [38] | - | Square-wave | Does not apply | 1000 | Does not apply | 400 V | Does not apply | 10% supramaximal (60–130 mA) |
| Maffiuletti et al., 2003 [72] | - | Rectangular | Single pulse | - | - | 400 V | Does not apply | Supramaximal (50–80 mA) |
| Hansen et al., 2003 [25] | - | Square | - | 800 | - | 8 ms inter-twitch interval | Does not apply | Doublet |
| Mela et al., 2001 [37] | - | Rectangular | 50* | 300 | 2 s/ not clear | - | Does not apply | Beneath the subject's pain threshold |
| Rassier, 2000 [39] | - | Square | 50 | 50 | 5 s / 5 s | - | 9 contractions | 50% MVC |
| Sacco et al., 1994 [61] | - | Rectangular | 30 | - | 15 s / 5 s | - | 6 contractions | 20% MVC |
| McNeal & Bake, 1988 [46] | | - | Single pulse | - | - | - | Does not apply | 60 mA |
| Fitch & McComas, 1985 [44] | - | Rectangular | 20 | 50 | 90s/0s | - | Single contraction | Supramaximal stimulus intensities |
| Fahey et al., 1985 [33] | asymmetrical bi-phasic | Square | 50 | Not informed | 10s/35s | 25 V; Pulse time to peak stimulus: 2 s | 60 contractions (15 min); fatigue not assessed | 42.6±4.4 mA (slowly increased to the highest level comfortably tolerated) |
| Marsh et al., 1981 [26] | - | Rectangular | 40* | 100 | - | - | Does not apply | 20% supramaximal |

IPI: Interpulse interval; bps: Bursts per second; MEIC: Maximal electrically induced contraction; VAS: Visual Analogue Scale; MVC: Maximal voluntary contraction; *maximum frequency/pulse width chosen among two or more options.

However, there was no significant effect (SMD: 1.48, CI 95%: -0.56, 3.53, p = 0.16) when comparing optimal and long muscle lengths during muscle belly stimulation (Fig 2B), drawn from a low-risk study [53] and a study that presented concerns, as the authors did not mention an adequate rest period between joint angle comparisons [42], yielding a moderate quality of evidence according to GRADE. For this stimulation type, no corresponding article was available for comparing other muscle lengths.

For the femoral nerve stimulation, all outcomes were graded as high-quality evidence. There was a significant effect of greater evoked torque favoring the optimal muscle length vs.

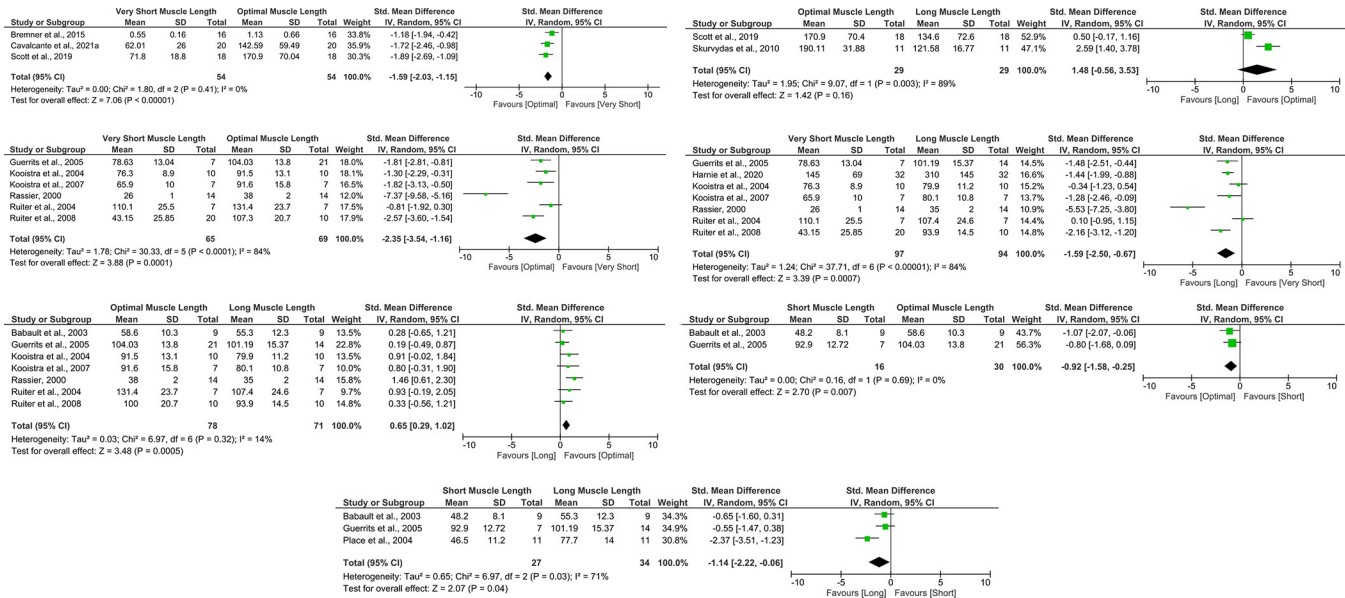

**Fig 2.** Meta-analyses of (A) very short muscle length vs. optimal muscle length for maximum evoked torque produced during quadriceps muscle motor point stimulation, (B) optimal muscle length vs. long muscle length for maximum evoked torque produced during quadriceps muscle motor point stimulation, (C) very short muscle length vs. optimal muscle length for maximum evoked torque produced during femoral nerve stimulation, (D) very short muscle length vs. long muscle length for maximum evoked torque produced during femoral nerve stimulation, (E) optimal muscle length vs. long muscle length for maximum evoked torque produced during femoral nerve stimulation, (F) short muscle length vs. optimal muscle length for maximum evoked torque produced during femoral nerve stimulation, and (G) short muscle length vs. long muscle length for maximum evoked torque produced during femoral nerve stimulation.

very short muscle length (SMD: -2.20, CI 95%: -3.42, -0.99, p < 0.001, Fig 2C) (16, 39, 52, 54–56), vs. short muscle length (SMD: -0.72, CI 95%: -1.43, -0.02, p = 0.04, Fig 2F) (16, 38), and vs. long muscle length (SMD: 0.63, CI 95%: 0.24, 1.02, p = 0.002, Fig 2E) (16, 38, 39, 52, 54–56). Moreover, there was a significant effect of greater evoked torque favoring the long muscle length vs. very short muscle length (SMD: -1.49, CI 95%: -2.40, -0.58, p < 0.001, Fig 2D) [16,39,52,54–57], and vs. short muscle length (SMD: -1.14, CI 95%: -2.22, -0.06, p = 0.04, Fig 2G) [16,38,43].

## NMES-induced contraction fatigue

Eight studies conducted an NMES-fatiguing protocol, six on the quadriceps femoris muscle (one of them through femoral nerve stimulation), and two on the tibialis anterior muscle. More highly flexed knee joint angles were more fatiguing for the quadriceps in five out of the six studies: 20° vs. 60° [14], 50° vs. 100° [28], 60° vs. 110° [58], 15° vs. 90° [59], and 30° vs 60° vs 90° [39], while in the investigation by Marion et al. [60], contraction fatigue was notably pronounced at the joint angle that exhibited the highest pre-fatigue force and the least contraction fatigue was observed at the joint angle characterized by a lower pre-fatigue force. In this instance, 65° of knee flexion demonstrated a greater pre-fatigue force compared to angles of 20° and 90°. However, Rassier [39] exhibited contradictory outcomes and reporting. The authors indicated a more substantial relative decline in torque at the most elongated position (90° compared to 60° and 30° of knee flexion), but the textual content consistently referred to greater contraction fatigue occurring at the most shortened muscle length. Regarding the tibialis anterior, findings suggested that NMES-induced contraction fatigue was more prominent at the optimal muscle length in contrast to the shortened position [44,61].

## Perceived discomfort

Six studies reported the perceived discomfort through a numeric scale. These studies used the Visual Analogue Scale (VAS), while one study also used the Verbal Rating Scale (VRS) [48]. Three studies utilized a VAS fixed value to achieve the maximum evoked contraction during quadriceps NMES: 7 out of 10 [53] and 8 out of 10 [11,14]. One study [47] found the current amplitude required to achieve 30% to 40% of the MVC at the more extended knee angle (15˚), and used the same current amplitude for the knee at 60˚ (which would produce a much lower percentage of the MVC for this angle) and compared the VAS. The authors found that, for 15˚ and 60˚, the VAS values were 32.42 mm (CI upper: 42.09; CI lower: 23.02) and 17.11 mm (CI upper: 27.34; CI lower: 7.09), respectively. One study, with a healthy older population [48], chose a fixed current amplitude (27 mA) for the stimulation of the common peroneal nerve to evoke isometric dorsiflexion (ankle angle not mentioned), to compare, among other variables, the perceived discomfort at three knee angles (0˚, 45˚, and 90˚). Pulse duration was gradually increased from 50 µs to 400 µs. At the highest pulse duration, no significant difference in perceived discomfort among knee angles was observed, as assessed using both the VAS and the VRS. In another study [62], a 20-minute NMES protocol was applied to the infraspinatus muscle. The participants were divided into four groups, based on joint angle and type of contraction (three for isometric and one for concentric contraction), as well as a control group. For the isometric contractions, three shoulder rotation angles were employed, with participants lying prone and shoulders abducted at 90˚: maximum internal rotation (82.5˚ ± 9.6˚), neutral rotation, and maximum external rotation (86.1˚ ± 13.5˚). The initial five minutes of muscle stimulation were utilized to progressively increase the current amplitude to the highest tolerable level for each subject (initial VAS not reported). This amplitude remained unchanged throughout the protocol. Following the protocol, no notable disparities were found among the groups (current amplitude [mA]: 83.5 ± 20.4, 85.0 ± 27.2, and 79.0 ± 20.4; VAS [mm]: 63.1 ± 19.0, 65.9 ± 18.8, and 59.8 ± 19.9 for the internal, neutral, and external rotation groups, respectively). A limitation of the study was the absence of force output measurement [62]. Regrettably, these studies could not be included in a meta-analysis due to these limitations.

## Strengthening NMES training

Among the included studies, only one [33] investigated the impacts of NMES on muscle strengthening as a function of muscle length. The young participants were trained at two joint angles: full knee extension (0˚) and a flexed position (65˚), in a supine position (hip angle unspecified). The outcomes revealed that NMES led to heightened isometric strength, with increases of 9.5% and 15.4% in the flexed knee group for males and females, respectively, and increases of 7.3% and 15.5% in the extended knee group for males and females, respectively, without difference between the groups. Notably, NMES exhibited greater effectiveness in enhancing isokinetic performance when the knee was flexed during treatment. Specifically, at the velocity of 30˚·s$^{-1}$ and 120·s$^{-1}$ for women, and 120·s$^{-1}$ for men, there was a greater improvement in the isokinetic torque.

## Discussion

This is the first systematic review to assess the effect of muscle length on maximum electrically induced torque, perceived discomfort, NMES-induced contraction fatigue, and strength following an NMES-strengthening program. We found that, for the quadriceps femoris muscle, the optimal muscle length for evoked torque was obtained at midrange (50˚-70˚) compared to short (regardless of stimulation method) and long muscle lengths (during nerve stimulation), reflecting its force-length relationship. In addition, the long length allowed greater evoked

torque than the short and very short muscle lengths. According to the GRADE recommendations [36], the quality of evidence was very high for most muscle length comparisons of the evoked torque, except optimal vs. long muscle length during muscle belly stimulation, which presented moderate quality of evidence. Other muscles, such as the ankle dorsiflexors and biceps brachialis, also exhibit an ideal (intermediate) length. However, for the ankle plantar flexors, longer lengths (either by manipulating the ankle or the knee joint angles) generated greater torque compared to short lengths, which also agrees with the torque-angle relationship for this muscle group that works in the ascending limb of the force-length relation [63].

When considering a given current amplitude, a shorter quadriceps muscle length seems to induce increased discomfort compared to the neutral position, although no definitive conclusions can be drawn regarding other muscle lengths and different muscles. NMES-induced contraction fatigue appears to be greater when the muscle length enables greater torque generation in the pre-fatigue condition. A protocol conducted at the ideal muscle length yielded greater quadriceps strength gains compared to a very short length, probably due to the greater mechanical work performed by the muscle at optimal length. Taken together, these findings suggest that prioritizing the optimal muscle length should be the initial choice for NMES interventions, avoiding shortened, low-demanding muscle lengths if clinically viable. However, indirect evidence from the included studies [62,64,65] and voluntary training elsewhere [66,67] point to the use of long quadriceps muscle lengths as a means of increasing mechanical stress during contraction and likely improving muscle force/mass and tendon properties, while the hamstrings may still benefit from training at shorter lengths [68]. Therefore, there are still many questions to be answered regarding NMES effects according to muscle length, with further research needed.

## Evoked torque

**'Very short' and 'short' vs. 'optimal' quadriceps femoris muscle length.**   While the force-length relationship is a well-established property of muscles fibers [21], this is the first meta-analysis confirming force disparities across multiple quadriceps femoris muscle lengths during electrical stimulation. As expected, an optimal muscle length surpasses very short or short muscle lengths in its capacity to evoke higher torque during NMES. This has significant implications for the utilization of NMES in both clinical and athletic contexts. Employing a shortened muscle length to facilitate a particular limb position (such as a fully extended knee) may likely constrain the potential benefits of NMES [31]. The selection of muscle length should be judicious, particularly in specific scenarios, such as cases involving limited/painful range of motion or with non-strengthening/hypertrophy goals.

**'Very short' and 'short' vs. 'long' quadriceps femoris muscle length.**   In the longmuscle length, NMES evoked higher torque when compared to both very short and short muscle lengths during nerve stimulation. It is noteworthy mentioning that the well-established knee extensor force-length relationship indicates potential alignment in force levels between short and long muscle lengths along the ascending and descending limbs of the curve [21]. However, it is important to consider that long muscle lengths tend to induce greater tensile stress [67], a factor favorable to enhance muscle (serial and parallel sarcomere) growth [66] and improve tendon mechanical properties [67]. This, in turn, lends support to the idea of choosing longer muscle lengths to accelerate the recovery of muscle mass. For instance, in the context of voluntary isometric knee extension exercises performed at short (43.1° ± 4.6° of knee flexion) and long (86.9° ± 6.5°) muscle lengths, the latter (long) can accelerate hypertrophy, while the former (short) yields greater strength enhancement at the training angle [66].

**'Optimal' vs. 'long' quadriceps femoris muscle length.** Two meta-analysis were conducted comparing optimal and long muscle lengths during two distinct methodologies: muscle belly and nerve stimulation. The results favored the optimal muscle length during nerve stimulation (seven studies) (Fig 2E), aligning with the expectations derived from the force-length relationship. Nevertheless, there was no difference during muscle stimulation (2 studies), which is likely attributed to the small number of studies and their heterogeneity. Consequently, opting for an optimal muscle length is likely to expedite progress in achieving strength gains. However, it is important to note that the aforementioned impact of tensile stress may render long muscle lengths more advantageous for other aspects related to muscle-tendon unit properties [67]. Moreover, long muscle lengths may lead to both parallel and serial sarcomerogenesis, which holds particular value in the treatment of various clinical conditions characterized by serial sarcomere loss (such as bed-rest periods, post-surgical interventions), followed by subsequent loss of functionality [66,69,70].

**Other muscle length comparisons for the quadriceps femoris.** Some studies could not be meta-analyzed [46,71]. Vissher et al. [71] compared the evoked torque through femoral nerve stimulation among three knee angles, finding a significantly lower peak twitch torque at 30˚ of knee flexion than at 65˚ and 90˚. However, the authors used a current amplitude of 100 mA for all tests, without mentioning if it evoked a maximal or supramaximal stimulus. Similarly, a pioneering study [46] standardized a 60 mA current amplitude for three different knee angles: 15˚, 45˚, and 75˚ during quadriceps femoris muscle belly stimulation. While similar current amplitude offers the advantage of facilitating the comparison of efficiency (i.e., the ratio of torque by current amplitude) [11], it does not inform about the maximum evoked torque capacity.

Maffiuletti et al. [72], employing femoral nerve stimulation, and Bampouras et al. [65], using muscle belly stimulation, both observed higher quadriceps evoked torque in the supine position compared to the seated position when the knee was fixed at 90˚. This suggests that the supine posture may position the biarticular rectus femoris muscle closer to its optimal length, a concept in line with the muscle's functional range [73]. However, when the knee is positioned at 60˚, this difference becomes less distinct and may or may not manifest [11]. An underlying explanation for these divergent outcomes might be the greater passive tension experienced by the knee extensors when the knee is flexed at 90˚, compared to the 60˚ flexion angle. These disparities could be attributed to the variance in the knee extensor passive tension under different knee angles. Future investigations hold the potential to elucidate whether manipulation of hip angles could confer benefits in the application of NMES to knee extensors.

Care should be taken concerning definitions of short, optimal, and long muscle length. For example, Fouré et al. [28] suggested that "rehabilitation training programs including electrically induced isometric contractions should be performed at short muscle lengths." This could misleadingly imply that short could be better than optimal in any NMES program. However, in the study, "short" was 50˚ and "long" was 100˚ of knee flexion, and 50˚ is closer to the angle range commonly reported as optimal (intermediate or midrange): 55˚ - 65˚ of knee flexion [11,38,71].

**Length comparisons for other muscle groups.** Our research retrieved studies that used other muscle length comparisons, which could not be meta-analyzed. Hali et al. [74] compared a shortened position (20˚ plantar flexed from neutral) and a lengthened position (20˚ dorsiflexed from neutral) for the triceps surae muscle and found greater peak twitch torque for the long muscle length (39.5 ± 12.5 vs 11.9 ± 4.8). While the aforementioned study does not include a neutral position for comparison, studies focusing on MVC have shown that greater plantar flexor torque is generated at lengthened muscle lengths compared to neutral positions

[63,75]. This indicates that the muscle-tendon unit and joint complex being examined differ from the knee extensor mechanism in terms of their response to muscle length, according to each muscle's mechanical properties (i.e., force-length relation). Similarly, the twitch and doublet (as well as the MVC) evoked torques for the plantar flexors increase when the knee is more extended, which lengthens the biarticular gastrocnemius, although, in their discussion, the authors seem to state the opposite to the information expressed in their results [76].

Mela et al. [37] stimulated the dorsiflexors through the stimulation of the deep peroneal nerve, while Marsh et al. [26] applied the stimulation directly over the tibialis anterior muscle belly. Specifically for Marsh et al. [26], who assessed several joint angles (from 30° of plantar flexion to 20° of dorsiflexion in steps of 5°), the greatest torque was obtained at 10° of plantar flexion. In the Mela et al. ([37] study, nerve stimulation applied to activate all four dorsiflexors resulted in an average evoked torque of 50% of MVC regardless of ankle joint angle. In healthy volunteers, evoked torque rarely reaches 100% of MVC [1]. Both authors, despite using different stimulation methods, consistently found that torque generation was greater in the plantar flexed (lengthened) position compared to the dorsiflexed (shortened) position, which also agrees whti the force-length relationship of the dorsiflexors, whose plateau occurs at 30° of plantar flexion [63]. This agreement emphasizes the influence of joint angle (and therefor muscle length) on torque production, irrespective of the specific details of the stimulation method.

Hansen et al. [25] assessed the force-angle relationship of elbow flexors during isometric contractions at various angles. The authors found that MVC, double twitch, and single twitch peak torques occurred at 90° (223.6 ± 56.3 N), 104° (223.6 ± 56.3 N), and 118° (223.6 ± 56.3 N), respectively. This indicated a rightward shift in the curves with submaximal force, possibly due to increased $Ca^{2+}$ sensitivity with muscle lengthening during submaximal contractions [77]. Future studies could the peak region of the force-angle relationship with NMES to more efficiently enhance elbow flexor strength. Additionally, Miyamoto and Oda [78] observed similar findings, with significantly higher torque at 120° (or 60°, considering full elbow extension as 0°) compared to more flexed angles. Future studies could also examine the impact of shoulder and forearm angles on the elbow flexor force-angle relationship during NMES.

## Contraction fatigability

A meta-analysis for this outcome could not be conducted due to discrepant methodologies across various studies. NMES-induced contraction fatigue appears to intensify when the muscle length enables greater torque generation in the initial (pre-fatigue) state, which may be explained by the increased metabolic demand of higher-intensity contractions, related to more actin-myosin cross-bridge formation [44]. Indeed, this is supported by the reduced oxygen consumption/metabolic rate at short muscle lengths [79]. However, also at shorter lengths, activity-dependent muscle fiber potentiation (enhanced submaximal contractility due to prior activity) is greater and may limit the detection of contraction fatigue [80]. Some studies that applied voluntary contraction-fatigue protocols are in agreement with these results [41], while others did not find significant differences in contraction fatigue according to muscle length [81]. These studies employ varying knee joint angles and contraction fatigue-protocol methodologies, limiting generalizability. For example, although the magnitude of force decline (fatigue itself) from isolated muscle-tendon units of rat medial gastrocnemius was also greater at longer muscle lengths, the rate of contraction fatigue (fatigability) was greater at short muscle lengths [82].

Notably, even with the increased contraction fatigue observed after NMES-fatigue protocols, optimal muscle lengths continue to yield greater torque compared to shorter lengths [14,63]. Therefore, the higher contraction fatigue observed at optimal length should not

preclude its preferential utilization in NMES strengthening programs, as it may still maintain the desired high muscle-tendon load [11,15]. However, new studies are needed to explore long-term outcomes. Additionally, for muscles spanning multiple joints, such as the quadriceps femoris, an added stretch, through hip extension, may also accelerate the onset of contraction fatigue and modulate strengthening adaptations [14], so this biomechanical aspect should not be overlooked when positioning the body parts. Finally, while the expectation that shorter muscle length generates less force, leading to reduced contraction fatigue compared to the optimal length, holds true for the quadriceps femoris, the same principle does not uniformly apply to the tibialis anterior muscle [61], opening the field for new studies to explore how different muscles respond acutely according to their fiber length and mechanical properties during NMES.

## Perceived discomfort

The assessment of perceived discomfort in the context of muscle length manipulation remains somewhat limited. We could not draw strong conclusions from the available studies. While it is often reported that shorter muscle lengths can induce painful cramp-like contractions, this hypothesis has not yet been subjected to rigorous testing [1]. Specifically, we found only one study evidencing that for the same current amplitude, greater discomfort is reported when quadriceps NMES is applied at 15˚ of knee flexion compared to 60˚. These results raise considerations for dynamic contractions with NMES, as care must be taken with excessive discomfort during the final range of motion. Interestingly, studies employing the VAS as a means to achieve maximum electrically induced contractions [11,53] have consistently yielded the same percentage of the maximum voluntary contraction. However, this may come at the cost of greater current amplitude applied at longer muscle lengths compared to shorter positions [11]. Thus, comprehensive understanding of the influence of muscle length on perceived discomfort leads to unavoidable questions, such as tracking the change in the localization of motor and sensory nerve branches under the electrodes with changes in joint angle, which may require meaningful strategies to optimize the clinical outcomes [83].

Finally, the study by Yanase et al. [62] did not include a means of monitoring torque, thereby restricting the comprehensive interpretation of the reported discomfort data. For comprehensive understanding of the impact of muscle length on NMES-induced perceived discomfort, the resulting torque (indicative of contraction intensity) also needs to be taken into account. Hence, future research should emphasize the development of torque control methodologies that are adaptable to diverse clinical settings and muscles, particularly in instances where dedicated equipment like a dynamometer chair is not readily available [31].

## Strengthening by NMES training

The effects of strength training with NMES are not yet fully understood. A solitary study [33] compared the results of a six-week NMES protocol at different knee joint angles (65˚ and 0˚). Participants were supine during the treatment protocol, but the hip angle was not mentioned. The authors found greater strength gains for the protocol at 65˚ of knee flexion during isokinetic tests. However, we partially answered our hypothesis because the improvement in muscle strength was equivalent for both groups during the isometric test at 65˚ (unfortunately, other angles were not tested), defying the angle-specificity adaptations (i.e., an increase in strength only in and close to the trained angle), which is not uncommon after isometric or partial range of motion training [67,68]. Interestingly, the participants [33] trained at full knee extension (i.e., at short knee extensor length), which would be assumed as being disadvantageous for torque generation and strengthening [11,31]. However, as the authors did not mention the hip

angle, if participants were fully supine in the full knee extension group, but with hip flexed in the knee flexed group, this adds a bias that tend to equalize muscle-tendon unit length and stiffness among positions [11]. No further studies are available for evaluating longer muscle lengths and hypertrophy. Despite the limited number of studies, a recent review [84] on the effects of NMES on knee post-surgery rehabilitation recommended positioning the patient at a 60˚ knee flexion angle if medically safe, justifying this advice by the anticipated greater torque in this position. Additional research on the adaptation of strengthening through NMES training with different joint positions is important to gain better understanding of its implementation in clinical settings.

Most NMES protocols primarily involve isometric contractions. While isometric contractions with a fully extended knee do not require specialized equipment, they may lead to reduced strength gains [31]. For precise control of joint angle and torque, a dynamometer chair is the optimal choice. However, access to these resources may be limited, particularly outside clinical facilities, such as in home care [31]. In resource-constrained clinical settings, particularly for bed-bound patients, a practical approach to managing joint angles during isometric contractions is by obtaining the desired knee flexion angle using a support under the lower limb (e.g., triangular wedge pillow) and ankle weights/restraints [85]. This approach, demonstrated by Toth et al. [86], effectively maintains the knee at a set angle. Another viable strategy involves seating the patient on a chair and using an adjustable strap to restrict knee extension, maintaining hip and knee angles at approximately 90˚ [2]. It is essential to note that to obtain high torque levels and, consequently, the most significant effect with these simple solutions, the current amplitude must be increased up to the maximum tolerated discomfort [84]. Additionally, to assess the quality of the contraction and monitor progress during the NMES program, visual inspection can be used, along with the option of manual resistance if needed.

## Limitations

The scope of our investigation was confined to evoked torque (non-potentiated), contraction fatigue (torque decline), discomfort, and chronic adaptation measures (strength and muscle mass gains) in response to NMES. However, certain included studies investigate other variables that may hold relevance for future reviews, such as the force-frequency relationship, differences in anatomical muscle-tendon-length characteristics (such as those seen in the plantar flexor), potentiated torque, M-wave, and the force-time integral. Moreover, the presence of multiple outcome measures introduces complexity and makes descriptive comparisons between studies challenging. Furthermore, the considerable variability across evaluations and clinical heterogeneity among studies precluded us from conducting more meta-analyses, consequently constraining this review to descriptive rather than quantitative comparisons. In addition, no previous meta-analyses have compared the different electrical stimulation techniques (e.g., single, doublet, tetanic, nerve vs muscle), and muscle length comparisons may be influenced by stimulation technique [87]. Lastly, despite conducting a comprehensive literature search across diverse databases, it is important to recognize that the search primarily targeted English-language journals, potentially overlooking studies in non-English publications and regional databases.

The included studies exhibited scores ranging from fair to good, and from 4 to 7 points on the PEDro scale. An assessment combining both the PEDro and RoB-CO frameworks reveals several concerns, particularly the absence of a clear delineation of methods to eliminate acute carryover effects, such as potentiation or contraction fatigue [39], since some studies compared joint angles on the same day, without justifying the allowed time for recovery between

joint angle tests (see "*Bias arising from period and carryover effects*" in S3 File). This is particularly pertinent when considering outcomes linked to maximum evoked torque. While examples of insufficient randomization were infrequent, they were evident in three studies [46,49,61]. Additionally, it is noteworthy that achieving blinding for both participants and assessors in studies involving joint angles can present inherent challenges. For forthcoming research endeavors, we suggest participant blinding pertaining to specific outcomes that have the potential to influence performance. Furthermore, it is essential to highlight that a significant number of studies did not incorporate a preliminary familiarization session, despite its potential to impact participant performance [88].

## Conclusion

In conclusion, optimal muscle length is key for maximizing torque generation during NMES. Longer muscle lengths also contribute to increased torque, while shorter lengths may result in greater discomfort. Tailoring joint angles to training goals can influence contraction fatigue. Quadriceps strength gains may be superior at the ideal muscle length compared to short muscle lengths, aiding muscle recovery. However, considering the limited NMES evidence, and conflicting findings after voluntary training, the use of short or long muscle lengths should be carefully selected based on diverse clinical factors, such as the available joint range of motion and the need for improvement in muscle-tendon unit properties. These findings are crucial for populations populations who present difficulty with volitional muscle contraction. Further research is needed to comprehensively assess the short- and long-term effects of varying muscle lengths on musculoskeletal adaptations after an NMES-strengthening program.

## Supporting information

**S1 File. Preferred Reporting Items for Systematic and Meta-Analyses Statement (PRISMA) checklist.**
(DOCX)

**S2 File. Search strategies according to database.**
(DOCX)

**S3 File. Risk of bias assessment according to the Revised Cochrane risk-of-bias tool for cross-over trials (RoB-CO).**
(PDF)

## Author Contributions

**Conceptualization:** Jonathan Galvão Tenório Cavalcante, Júlia Aguillar Ivo Bastos, Marco Aurélio Vaz, Nicolas Babault, João Luiz Quagliotti Durigan.

**Data curation:** Jonathan Galvão Tenório Cavalcante, Victor Hugo de Souza Ribeiro.

**Formal analysis:** Jonathan Galvão Tenório Cavalcante, Victor Hugo de Souza Ribeiro.

**Funding acquisition:** Jonathan Galvão Tenório Cavalcante, Rita de Cássia Marqueti, João Luiz Quagliotti Durigan.

**Investigation:** Jonathan Galvão Tenório Cavalcante, Victor Hugo de Souza Ribeiro, Júlia Aguillar Ivo Bastos.

**Methodology:** Jonathan Galvão Tenório Cavalcante, Victor Hugo de Souza Ribeiro, Rita de Cássia Marqueti, Isabel de Almeida Paz, Júlia Aguillar Ivo Bastos, Marco Aurélio Vaz, Nicolas Babault, João Luiz Quagliotti Durigan.

**Project administration:** Jonathan Galvão Tenório Cavalcante, Victor Hugo de Souza Ribeiro, Marco Aurélio Vaz, Nicolas Babault, João Luiz Quagliotti Durigan.

**Resources:** Jonathan Galvão Tenório Cavalcante, João Luiz Quagliotti Durigan.

**Software:** Jonathan Galvão Tenório Cavalcante, Victor Hugo de Souza Ribeiro, Júlia Aguillar Ivo Bastos.

**Supervision:** Jonathan Galvão Tenório Cavalcante, Rita de Cássia Marqueti, Marco Aurélio Vaz, Nicolas Babault, João Luiz Quagliotti Durigan.

**Validation:** Jonathan Galvão Tenório Cavalcante, Victor Hugo de Souza Ribeiro, Rita de Cássia Marqueti, Isabel de Almeida Paz, Júlia Aguillar Ivo Bastos, Marco Aurélio Vaz, Nicolas Babault, João Luiz Quagliotti Durigan.

**Visualization:** Jonathan Galvão Tenório Cavalcante, Victor Hugo de Souza Ribeiro, Rita de Cássia Marqueti, Isabel de Almeida Paz, Júlia Aguillar Ivo Bastos, Marco Aurélio Vaz, Nicolas Babault, João Luiz Quagliotti Durigan.

**Writing – original draft:** Jonathan Galvão Tenório Cavalcante, João Luiz Quagliotti Durigan.

**Writing – review & editing:** Jonathan Galvão Tenório Cavalcante, Victor Hugo de Souza Ribeiro, Rita de Cássia Marqueti, Isabel de Almeida Paz, Júlia Aguillar Ivo Bastos, Marco Aurélio Vaz, Nicolas Babault.

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
