## [Decision Letter · Decision Letter 0]

16 Feb 2024

PONE-D-23-35590Effect of muscle length on maximum evoked torque, discomfort, fatigue, and strength adaptations during electrical stimulation in adult populations: a systematic reviewPLOS ONE

Dear Dr. Cavalcante,

Thank you for submitting your manuscript to PLOS ONE. After careful consideration, we feel that it has merit but does not fully meet PLOS ONE’s publication criteria as it currently stands. Therefore, we invite you to submit a revised version of the manuscript that addresses the points raised during the review process.

I would encourage you to consider carefully the points made by both reviewers, especially around clarity re the number of studies, the specific inclusion / exclusion criteria etc. You will see the 2nd reviewer has made several recommendations and I think they will improve the clarity of the manuscript. 

We look forward to receiving your revised manuscript.

Kind regards,

Theodoros M. Bampouras

Academic Editor

PLOS ONE

Journal Requirements:

3. "We note that the grant information you provided in the ‘Funding Information’ and ‘Financial Disclosure’ sections do not match. 

4. Thank you for stating the following in the Acknowledgments Section of your manuscript: "This study was financed in part by the Coordenação de Aperfeiçoamento de Pessoal de Nível Superior, Brazil (CAPES), Finance Code 001, Fundação de Apoio à Pesquisa do Distrito Federal (FAPDF) (grant numbers 00193-00001615/2023-00, 00193-00002357/2022-90, 00193.00001222/2021-26, 00193-00001261/2021-23), and the National Council for Scientific and Technological Development (CNPq; process number 310269/2021). M.A.V. AND J.L.Q.D are recipient of a research grant from the National Council for Scientific and Technological Development."

Please remove any funding-related text from the manuscript and let us know how you would like to update your Funding Statement. Currently, your Funding Statement reads as follows: "J.G.T.C received a grant by Fundação de Apoio à Pesquisa do Distrito Federal (FAPDF) (grant number 00193-00001615/2023-00) for paying english language review and publication fee for the present study. FAPDF is a public research support foundation.

" ext-link-type="uri" xlink:type="simple">https://www.fap.df.gov.br/"

Reviewers' comments:

Reviewer's Responses to Questions

**Comments to the Author**

1. Is the manuscript technically sound, and do the data support the conclusions?

Reviewer #1: Yes

Reviewer #2: Yes

2. Has the statistical analysis been performed appropriately and rigorously? 

Reviewer #1: Yes

Reviewer #2: Yes

3. Have the authors made all data underlying the findings in their manuscript fully available?

Reviewer #1: Yes

Reviewer #2: Yes

4. Is the manuscript presented in an intelligible fashion and written in standard English?

Reviewer #1: Yes

Reviewer #2: No

5. Review Comments to the Author

Reviewer #1: This systematic review paper analyses the effect of muscle length on parameters such as evoked torque, fatigue response/fatigability, discomfort, and strength and muscle size changes in studies using neuromuscular electrical stimulation. Systematic analyses were done for all the abovementioned parameters, while only evoked torque was eligible for a meta-analysis due to the heterogenicity and scarcity of studies on the other parameters. The review shows that there is an optimal muscle length at which the highest torque can be evoked. Conversely, shorter and longer muscle lengths provide lower torque levels, much in line with previous studies of the muscle force-length relationship. The observed fatigue appears to follow the highest torque production, so that the muscle length (joint angle) that provides the highest torque also gives the most pronounced fatigue response. For discomfort, there is no clear pattern, although some evidence suggests that electrically evoked contractions at shorter muscle lengths may be less comfortable. Only one study investigated the effect of muscle length on training adaptations, showing that training with knee flexion was superior to an extended knee for strength adaptations. The review concludes that training interventions should aim to use the optimal muscle length as opposed to short muscle lengths, unless specific factors suggested otherwise. Meanwhile, training with longer muscle lengths may also be effective for stimulating muscle hypertrophy, but more evidence is needed to draw firm conclusions.

General comments: There are no line numbers for the first half of the manuscript. The line numbers start in the results section. I have deleted the present line numbers and inserted them again starting from the title page (title starting in line 1) and onwards.

The main body of evidence in this manuscript revolves around the investigation of evoked torque at different muscle lengths / joint angles. The conclusion very closely follows knowledge from the force-length relationship. Please acknowledge that this knowledge would be expected, although some practical aspects of testing muscle length in vivo might have had an influence on results from human muscle. For the other parameters, meta-analyses were precluded. Please be very specific on what issues need to be addressed in future studies in your discussion section, and also be specific with scenarios/patient groups where NMES provide rehabilitation opportunities that cannot be replaced by other rehabilitation strategies. This would help clinicians and future studies.

Even though not explicitly stated in title, abstract or inclusion criteria, all studies included use isometric contractions to compare muscle lengths. Please make sure to clarify that only isometric contractions were used for analyses. Otherwise, studies such as PMID: 31423755 could be included for the analysis of training outcomes.

Specific comments

L91-93: This statement about optimization of NMES sounds like it’s mostly related to the analyses of training outcomes and discomfort, which are likely the most clinically relevant analyses, but which are also the areas with the least/weakest data. Please rephrase and make sure the statement also relates well to the analysis of evoked torque and fatigue.

L94-107: This section currently lacks a concluding remark or point.

L109: Is this narrative review the only instance where short muscle lengths are mentioned as being less comfortable, and without a reference to an original study?

L115: What is meant by “progress towards longer muscle length” in this context?

L127: Throughout the article, please be very clear on what is meant by “fatigue” and “fatigability”, which is not the same thing and may have different definitions in different articles. It would be helpful if you also specified the fatigue/fatigability definitions used by your analyzed studies.

L131: This statement covers the importance of the analyses of training outcomes. See my first specific comment and make sure the statements cover all your outcomes fairly.

L172: Not all readers will be aware of what ‘grey literature’ means. Please specify.

L176-182: As mentioned in the general comments, all studies seem to use isometric contractions to compare muscle lengths on the different parameters. Please make sure to specify the reason either in your inclusion criteria or otherwise. Again, I don’t see why e.g. PMID: 31423755 does not live up to the inclusion criteria.

L198: Do you think that the optimal length for torque production during twitch, doublet, or continuous stimulation is the same? Please comment

L202: Replace “where” with “was” or “were”.

L217: Make sub header “Quality of Evidence” italic/cursive

L253: Some studies find ~70 degrees knee flexion to be optimal for MVC torque production, e.g. DOI: 10.1016/j.jbiomech.2004.02.005. Please acknowledge that the range for optimal length may be wider than you have currently written. Also, ref 33 does not include quadriceps strength measurements, but dorsiflexors, as I understand it.

L295: Correct “3” to “three”

L312: Please elaborate briefly on what ‘extra forces’ means, as some readers may not be familiar with the manuscript by Frigon et al.

L321: Please briefly describe what a Russian current is.

L328: There are some inconsistencies regarding the use of numerals or numbers written with words. Please keep consistent. In this case change for ”23”, probably

L339-341: How were the measurements combined? As an average for the angles in each category? Please comment.

L372: If this is only true in the 4 out of 6 cases, it should be specified that the remaining studies did not reach the same conclusion.

L372-375: Do you think that the for the fatigability, the muscle length itself plays a role, or do you think that it is simply the resulting force/torque that decides the fatigue development? Perhaps something you could discuss in your discussion section.

L418-425: The only study investigating the effect of training at various muscle lengths only use a (seemingly) very short quadriceps length and a flexed position. As the short length seems to be very short,

L430-432: The fact that the optimal muscle length yields greater contraction force than short or long muscle lengths is rather self-explanatory. Please rephrase. Perhaps that the force-length relationship is apparent in electrically evoked contractions in the quadriceps femoris in vivo.

L433: Long lengths allowed greater torque than and AND very short lengths as I understand it.

L437: replace “plantarflexors” with “plantar flexors”

L447-450: Here you are very conservative when discussing the potential usefulness of training at long muscle lengths. It would be better if you could be clear on the potential benefits/drawbacks of training at long muscle lengths and what still needs to be investigated to make more firm conclusions.

L454-458: I believe the finding of the capacity of the quadriceps to produces more torque at optimal muscle lengths is not quite as novel as it is made out to sound here.

L463-466: Should this part about short vs. long muscle lengths not be in the following section instead of in the section about short vs. optimal muscle lengths? Depending on how you define “long” of course, and whether you want to call 86.9 degrees long or optimal.

L476-477: This sentence end abruptly. Aligning with long-term goals of what?

L499: Please also explain why reference 57 could not be a part of a meta-analysis.

L538: This sentence is not clear. Did Marsh et al. state that they didn’t reach MVC torque levels because of a lack of synergist activation during evoked contractions? And what are the implications for the conclusions of this section if they misinterpreted their findings in such a way?

L553: What is meant by ”submaximal curves” in this context?

L574-577: Please acknowledge that studies still need to confirm that training at optimal muscle length and higher torque production leads to greater improvements in strength and/or muscle hypertrophy compared to lower torques/suboptimal muscle lengths.

L574-581: You cannot equate accelerated fatigue and an increased muscle hypertrophic response without a reference underpinning this statement. Muscle work can be very fatiguing without leading to muscle hypertrophy.

L599-600: By “it lacks”, do you mean the study of Yanase et al., or the method that they used?

L611-612: As you only have one study in this analysis, please be more specific and include the increase in strength using numbers or %, and with some statistical indication of certainty, such as 95 % confidence intervals or similar, as provided in the reference. This information could also be inserted in the results section instead.

L613: ”also” is redundant.

L616-617: Please elaborate on how they reach this conclusion and how it aligns with the results of the present study.

L624: It is not clear from the description how the leg is secured and positioned. Please elaborate

L630: Are you saying that the current apmplitude matters more than the precise muscle length?

L647: Insert full stop after sentence.

L651: What is meant by “carryover effects” in this context?

L657-659: “…studies have not incorporated a preliminary familiarization session, despite its potential to impact participants' performance”. Insert full stop after sentence. Also, could add doi: 10.1016/j.apmr.2022.09.004. to substantiate this statement.

L663-664: “Quadriceps strength gains are superior at the ideal muscle length, aiding muscle recovery.“ This statement is quite strong from the limited evidence from training studies. Please make it more moderate as there is a scarcity of studies and actual angles tested for training outcomes.

L723-724: Make “TRANSLATIONAL SPORTS MEDICINE” lower case

L843-846: Make ref. 55 lower case. 

Table 1: The Y/N denotation of PEDro scale evaluation is not easily read. Could a color code be useful here? E.g. white for Y, grey for N. Just a suggestion. Where is the table legend?

Table 1: Both the PEDro table and the following study characteristics table are called “Table 1”. Please rename. Also, for the characterization table:

- In the electrode placement of Miyamoto Oda, 2005: Replace “biceps” with “biceps”

- In the study by Hansen et al. 2003, replace “braquialis” with “brachialis”

- In the study by Mela et al., 2001, replace “Cachode” with “Cathode”

- In the study by Sacco et al.,1994: Replaces “TIbialis” with “Tibialis”

Table 2: In the studies by Cavalcante et al., 2022: The abbreviation “EVA” has not been defined anywhere.

Fig 1. from the last step before the screen steps, there are listed ”From database search: n = 826” and ”From other sources: n = 6”, so a total of 832 articles. But in the next step, there are only 826 titles screened. Are the 6 articles from other sources the same as in the last step (”included”)? Also, for the 826 titles screened, 761 records are excluded, giving 65 titles left. The figure shows only 63 titles left. Please check your flow chart for errors in the number of excluded and included articles for each step.

Reviewer #2: Please provide a list of better key words related to the topic of your manuscript. MeSH terms are recommended.

Please list reasons for excluding 6 studies that "reports assessed for eligibility in the PRISMA figure. Make sure the numbers in each box add up for example after excluding 761 records you may had 65 studies not 63 as indicated in PRISMA.

Please correct the typo on last line of page 7. ...when the data where not available,....

Discussion: The authors have effectively crafted this section, captivating the reader with a wealth of knowledge. To enhance the informative value, it would be beneficial for the authors to explore the impacts of anatomical muscle-tendon-length characteristics, such as the various tendons length of Plantar Flexors to their contractile bulk. This comparison could shed light on the ideal joint angles reported for maximum force production across the examined protocols.

6. PLOS authors have the option to publish the peer review history of their article (what does this mean?). If published, this will include your full peer review and any attached files.

Reviewer #1: No

Reviewer #2: No

---

## [Author Response · Author response to Decision Letter 0]

1 Apr 2024

Thank you for the opportunity to answer the reviewers and upload our revised manuscript.

The comments from the reviewers were highly insightful and enabled us to improve the quality of our manuscript. We have made several changes to address the suggestions. A point-by-point response to the reviewers is provided bellowe, and the marked-up, and unmarked versions of the manuscript were uploaded, as well the necessary adjustments in the supplemental files. 

Thank you again for your time and effort in considering this manuscript for publication.

---

## [Decision Letter · Decision Letter 1]

1 May 2024

PONE-D-23-35590R1Effect of muscle length on maximum evoked torque, discomfort, contraction fatigue, and strength adaptations during electrical stimulation in adult populations: a systematic reviewPLOS ONE

Dear Dr. Cavalcante,

Thank you for submitting your manuscript to PLOS ONE. After careful consideration, we feel that it has merit but does not fully meet PLOS ONE’s publication criteria as it currently stands. Therefore, we invite you to submit a revised version of the manuscript that addresses the points raised during the review process.

There reviewer is happy with the amendments and have requested some minor corrections. Please address those and proof-read teh manuscript for any additional passages that require corrections.  

We look forward to receiving your revised manuscript.

Kind regards,

Theodoros M. Bampouras

Academic Editor

PLOS ONE

Journal Requirements:

Reviewers' comments:

Reviewer's Responses to Questions

**Comments to the Author**

1. If the authors have adequately addressed your comments raised in a previous round of review and you feel that this manuscript is now acceptable for publication, you may indicate that here to bypass the “Comments to the Author” section, enter your conflict of interest statement in the “Confidential to Editor” section, and submit your "Accept" recommendation.

Reviewer #1: (No Response)

2. Is the manuscript technically sound, and do the data support the conclusions?

Reviewer #1: Yes

3. Has the statistical analysis been performed appropriately and rigorously? 

Reviewer #1: Yes

4. Have the authors made all data underlying the findings in their manuscript fully available?

Reviewer #1: Yes

5. Is the manuscript presented in an intelligible fashion and written in standard English?

Reviewer #1: Yes

6. Review Comments to the Author

Reviewer #1: I thank the authors for their corrections and further considerations on their manuscript. I have only a few minor comments to the current manuscript:

Line numbers of clean document (without tracked changes):

l. 87: “As such, the optimal muscle length (or ideal, or preferable) for NMES,..”

It would be preferable if you could settle on one of the words and delete the parenthesis.

l. 689-692 “… and muscle length comparisons may be influenced by stimulation technique.”

Thank you for this correction. Please add a reference to this statement, e.g. PMID: 12450066

7. PLOS authors have the option to publish the peer review history of their article (what does this mean?). If published, this will include your full peer review and any attached files.

Reviewer #1: No

---

## [Author Response · Author response to Decision Letter 1]

2 May 2024

TO THE ACADEMIC EDITOR

“”

Reply: Thank you for the information. No change is necessary.

“If applicable, we recommend that you deposit your laboratory protocols in protocols.io to enhance the reproducibility of your results. Protocols.io assigns your protocol its own identifier (DOI) so that it can be cited independently in the future. For instructions see: https://journals.plos.org/plosone/s/submission-guidelines#loc-laboratory-protocols. Additionally, PLOS ONE offers an option for publishing peer-reviewed Lab Protocol articles, which describe protocols hosted on protocols.io. Read more information on sharing protocols at https://plos.org/protocols?utm_medium=editorial- emailutm_source=authorlettersutm_ campaign=protocols.”

Reply: Thank you. Not applicable.

“Please review your reference list to ensure that it is complete and correct. If you have cited papers that have been retracted, please include the rationale for doing so in the manuscript text, or remove these references and replace them with relevant current references. Any changes to the reference list should be mentioned in the rebuttal letter that accompanies your revised manuscript. If you need to cite a retracted article, indicate the article’s retracted status in the References list and also include a citation and full reference for the retraction notice.”

Reply: Thank you for pointing to this relevant double check in our reference list. We did not cite retracted papers. We added a new reference based on reviewer’s suggestion: “87. Rassier DE, MacIntosh BR. Length-dependent twitch contractile characteristics of skeletal muscle. Can J Physiol Pharmacol. 2002;80(10):993-1000.”

“While revising your submission, please upload your figure files to the Preflight Analysis and Conversion Engine (PACE) digital diagnostic tool, https://pacev2.apexcovantage.com/. PACE helps ensure that figures meet PLOS requirements. To use PACE, you must first register as a user. Registration is free. Then, login and navigate to the UPLOAD tab, where you will find detailed instructions on how to use the tool. If you encounter any issues or have any questions when using PACE, please email PLOS at figures@plos.org. Please note that Supporting Information files do not need this step.”

Reply: Thank you for the helpful tool. The last upload of our figures (during the first review) was performed after using PACE to make the files meet PLOS requirements.

 

REVIEWER 1

Dear Reviewer, thank you again for your valuable comments and for appreciating our first rebuttal letter. We are confident that the manuscript has significantly improved based on your suggestions. Your comments have been addressed point-by-point and are provided below.

Reviewer #1: “I thank the authors for their corrections and further considerations on their manuscript. I have only a few minor comments to the current manuscript:

Line numbers of clean document (without tracked changes):

l. 87: “As such, the optimal muscle length (or ideal, or preferable) for NMES,..”

It would be preferable if you could settle on one of the words and delete the parenthesis.

Answer: Thank you for mentioning this necessary adjustment. We have adjusted the text accordingly: “As such, the optimal muscle length (or ideal, or preferable) for NMES…”

l. 689-692 “… and muscle length comparisons may be influenced by stimulation technique.”

Thank you for this correction. Please add a reference to this statement, e.g. PMID: 12450066

Answer: Thank you for the suggestion. We agree that a sound reference was missing and we considered appropriate to add the suggested reference: “and muscle length comparisons may be influenced by stimulation technique (87).”

87. Rassier DE, MacIntosh BR. Length-dependent twitch contractile characteristics of skeletal muscle. Can J Physiol Pharmacol. 2002;80(10):993-1000.

---

## [Editor Report · Decision Letter 2]

8 May 2024

Effect of muscle length on maximum evoked torque, discomfort, contraction fatigue, and strength adaptations during electrical stimulation in adult populations: a systematic review

PONE-D-23-35590R2

Dear Dr. Cavalcante,

We’re pleased to inform you that your manuscript has been judged scientifically suitable for publication and will be formally accepted for publication once it meets all outstanding technical requirements.

Kind regards,

Theodoros M. Bampouras

Academic Editor

PLOS ONE
---

## [Editor Report · Acceptance letter]

15 May 2024

PONE-D-23-35590R2 

PLOS ONE

Dear Dr. Cavalcante, 

I'm pleased to inform you that your manuscript has been deemed suitable for publication in PLOS ONE. Congratulations! Your manuscript is now being handed over to our production team.

Kind regards, 

on behalf of

Dr. Theodoros M. Bampouras 

Academic Editor

PLOS ONE